

# Long-term reconstruction of satellite-based precipitation, soil moisture, and snow water equivalent in China

Wencong Yang[1, 2], Hanbo Yang[1,2], Changming Li[1,2], Taihua Wang[1,2], Ziwei Liu[1,2], Qingfang Hu[3], and Dawen Yang[1,2]

[1] Department of Hydraulic Engineering, Tsinghua University, Beijing 100084, China.
   [2] State Key Laboratory of Hydro-Science and Engineering, Tsinghua University, Beijing, 100084, China.
   [3] State Key Laboratory of Hydrology-Water Resources and Hydraulic Engineering & Science, Nanjing Hydraulic Research Institute, Nanjing 210029, China.

*Correspondence to*: Hanbo Yang (yanghanbo@tsinghua.edu.cn)

**Abstract.** A long-term high-resolution national dataset of precipitation (*P*), soil moisture (SM), and snow water equivalent (SWE) is necessary for predicting floods and droughts and assessing the impacts of climate change on streamflow in China. Current long-term daily or sub-daily datasets of *P*, SM, and SWE are limited by a coarse spatial resolution, the lack of local correction, or the lack of direct calibration. In this study, we produced a daily 0.1° dataset of *P*, SM, and SWE in 1981-2017 across China using global background data and local onsite data as forcing input and satellite-based data as reconstruction

benchmarks. Long-term global 0.1° and local 0.25° *P* data are merged to reconstruct the *P* from the short-term 0.1° China Merged Precipitation Analysis (CMPA) using a stacking machine learning model. Long-term SM data are reconstructed by the HBV hydrological model with SM calibrated by Soil Moisture Active Passive Level 4 (SMAP-L4) data. Long-term SWE data are also reconstructed by the HBV hydrological model with SWE calibrated by the national satellite-based snow depth dataset in China (Che and Dai, 2015) and the Moderate Resolution Imaging Spectroradiometer (MODIS) snow cover data. For

all grids at the national and daily scale, the median Kling-Gupta Efficiency (KGE) of the reconstructed *P* and SM are 0.68 and 0.61 respectively. For grids in two snow-rich regions at the daily scale, the median KGEs of the reconstructed SWE are 0.55 and -2.41 in the Songhua and Liaohe Basin and the Continental Basin respectively. Generally, the reconstruction dataset performs better in southern and eastern China than in northern and western China for *P* and SM, and performs better in northeast China than other regions for SWE. As the first long-term 0.1° daily dataset of *P*, SM, and SWE that combines

information from local observations and satellite-based data benchmarks, this reconstruction product is valuable for future national investigations of hydrological processes.

## 1 Introduction

A long-term national terrestrial hydrological dataset with high spatial-temporal resolutions can be used in many hydrological applications such as: exploring the controls of rainfall-runoff events (Tarasova et al., 2020; Yang et al., 2020; Stein et al.,

2021), predicting floods and droughts (Van Steenbergen and Willems, 2013; Reager et al., 2014; Abelen et al., 2015), and assessing the impacts of climate change on streamflow and floods (Sharma and Wasko, 2018; Blöschl et al., 2019; Li et al, 2019). As key variables in the hydrological cycle, precipitation (*P*), soil moisture (SM), and snow water equivalent (SWE)



generate riverine runoff and determine the wetness states of the basins. Although long-term (at least 30 years) daily $P$, SM, and SWE can be obtained from many data products in China, these products suffer from a coarse spatial resolution and a lack

of local information.

## 1.1 Limitations of long-term daily or sub-daily precipitation data in China

There are many long-term global precipitation data with a temporal resolution within one day and a spatial resolution within $0.1°$. For example, two popular datasets are the Multi-Source Weighted-Ensemble Precipitation (MSWEP; Beck et al., 2019) and the hourly $0.1°$ dataset ERA5-land (Muñoz-Sabater et al., 2021). MSWEP is a 3-hourly $0.1°$ dataset that begins in 1979

and merges multiple sources including gauge stations, remote sensing observations, and reanalysis data. ERA5-land is an hourly $0.1°$ reanalysis dataset that begins in 1981. Those global datasets have insufficient information of data from rain gauge stations in China, which leads to limited performance. Two local precipitation datasets are widely used in China. The first one is the China Gauge-based Daily Precipitation Analysis (CGDPA; Shen and Xiong, 2016), which is interpolated from the daily data back to 1960 in approximately 2400 ground stations. The key limitation of CGDPA is its coarse spatial resolution of $0.25°$.

The second one is the China Meteorological Forcing Dataset (CMFD; He et al., 2020), which is a 3-hourly $0.1°$ dataset in 1979-2018 using approximately 700 ground stations to correct the Global Land Data Assimilation System (GLDAS; Rodell et al., 2004) and Tropical Rainfall Measuring Mission (TRMM; Huffman et al., 2007) precipitation background data. CMFD does not take full advantage of available precipitation information since better background data (e.g., MSWEP and ERA5-land) and more ground station data are available.

## 1.2 Limitations of long-term daily or sub-daily soil moisture data in China

Remote sensing and reanalysis data are two common types of global soil moisture data. The European Space Agency's Climate Change Initiative (ESA-CCI; Dorigo et al., 2017) for soil moisture is a global satellite-monitored dataset that begins in 1979. However, ESA-CCI only measures surface soil moisture up to 5 cm depth in a coarse spatial resolution of 25 km. Global reanalysis data, e.g., ERA5-land, can provide soil moisture data in deeper soil layers in a high spatial resolution. However,

global reanalysis data miss observational information of soil moisture, and they simulate soil moisture using global forcing data which lack local corrections. Many hydrologic fluxes and states datasets provide simulated soil moisture data at a national scale. For example, the 3-hourly $0.25°$ dataset based on the VIC model created by Zhang et al. (2014), the daily $0.25°$ dataset based on the VIC model created by Miao and Wang (2020), and the daily $0.0625°$ dataset based on the VIC model created by Zhu et al. (2021). However, all these national-scale soil moisture data are simulated by a hydrological model calibrated by

only observed streamflow data. Therefore, the lack of direct calibration in observed soil moisture causes uncertain accuracy.

## 1.3 Limitations of long-term daily or sub-daily snow water equivalent data in China

Similar to soil moisture, remote sensing data and reanalysis data are two common types of global data products for snow water equivalent. GlobSnow (Luojus et al., 2021) is a global daily snow water equivalent dataset that assimilates satellite radiometer





data and ground snow depth observations. To promote local applications, Che and Dai (2015) developed a national satellite-
based snow depth dataset in China (abbreviated as SD-CN hereafter). Both GlobSnow and SD-CN begin in 1979 with a coarse
spatial resolution of 25 km. Global reanalysis data such as ERA5-land, as we stated before, use global forcing input with
limited local information. The snow water equivalent data from the national hydrologic fluxes and states datasets (Zhang et
al., 2014; Miao and Wang, 2020) have the same problem as soil moisture data, i.e., they are not directly calibrated using any
snow observations.

**1.4 Objectives**

We aim to use both global and local forcing data as input and satellite-based data as model training or calibration targets to
reconstruct historical hydrological variables. This kind of reconstruction is promising in producing long-term high-resolution
datasets with the following advantages. First, many satellite-based data have high spatial resolutions. For example, the $0.1°$
China Merged Precipitation Analysis (CMPA; Shen et al., 2014; Shen et al., 2018) from 2008 for precipitation, the 9 km Soil
Moisture Active Passive level 4 data (SMAP-L4; Reichle et al., 2019) from 2015 for root zone soil moisture, and the 500 m
Moderate Resolution Imaging Spectroradiometer (MODIS; Hall et al., 2002) from 2000 for snow cover. Second, combining
global and local forcing data as input not only increases local reconstruction accuracy, but also produces a physically consistent
dataset of the combination of $P$, SM, and SWE, since they are the hydrological fluxes and states from the same modeling
system during the reconstructions.

In this study, we produced a daily $0.1°$ dataset of $P$, SM, and SWE in 1981-2017 in China. For $P$, we merged CGDPA and
MSWEP to reconstruct the $P$ from CMPA using machine learning techniques; for SM, we used the reconstructed $P$ to drive a
hydrological model to reconstruct SM from SMAP level 4; for SWE, we also used the reconstructed $P$ to drive a hydrological
model calibrated by SD-CN and MODIS snow cover data to reconstruct multiple snow-related variables, e.g., snowfall,
snowmelt, and SWE. This is the first long-term (at least 30 years) $0.1°$ daily dataset of $P$, SM, and SWE that combines local
information and satellite-based data.

**2 Data**

This study used two categories of data. The first category includes the forcing and auxiliary data, which are the input of the
reconstruction methods. The second category includes the validation data of $P$, SM, and SWE, which are the evaluation
benchmarks of the reconstruction methods.

**2.1 Forcing and auxiliary data**

Information about the forcing data and auxiliary data are listed in Table 1. Precipitation data include the China Gauge-based
Daily Precipitation Analysis (CGDPA; Shen and Xiong, 2016) and the Multi-Source Weighted-Ensemble Precipitation
(MSWEP version 2.2; Beck et al., 2019). The daily $0.25°$ CGDPA data are produced using a spatial interpolation of



observations from approximately 2400 ground rain gauge stations. The 3-hourly 0.1° MSWEP data are produced by optimally
merging a range of gauge station, satellite, and reanalysis datasets. Air temperature ($T$) data include the observations from
approximately 2400 ground stations provided by the Chinese Meteorological Administration and ERA5-land 2 m temperature
(Muñoz-Sabater et al., 2021). Note that the number of available ground stations for $T$ is around 800 before 1988. Net radiation
($R_n$) data are from ERA5-land (Muñoz-Sabater et al., 2021). Elevation (Elev) data are from MERIT-Hydro (Yamazaki et al.,
2019). Leaf area index (LAI) data are from the Global Land Surface Satellite (GLASS; Liang et al., 2021) dataset.

**Table 1. Sources of forcing and auxiliary data.**

| Variable | Dataset | Spatial resolution | Temporal resolution | Temporal coverage | Reference |
|---|---|---|---|---|---|
| Precipitation | CGDPA | 0.25° | daily | 1960-2020 | Shen and Xiong, 2016 |
| Precipitation | MSWEP | 0.1° | 3-hourly | 1979-2017 | Beck et al., 2019 |
| Air temperature | Stations | — | daily | 1960-2019 | Chinese Meteorological Administration |
| Air temperature | ERA5-land | 0.1° | hourly | 1981-now | Muñoz-Sabater et al., 2021 |
| Net radiation | ERA5-land | 0.1° | hourly | 1981-now | Muñoz-Sabater et al., 2021 |
| Elevation | MERIT-Hydro | 90 m | — | — | Yamazaki et al., 2019 |
| Leaf area index | GLASS | 0.05° | 8-day | 1981-2017 | Liang et al., 2021 |

## 2.2 Validation data

Information about the validation data is listed in Table 2. All following satellite-based data provide direct or indirect
measurements of the variables to be reconstructed over a large spatial extent. $P$ data include the China Merged Precipitation
Analysis (CMPA; Shen et al., 2014) and its successor, CMPA_1km (Shen et al., 2018). CMPA merge more than 30000
automatic weather stations with the Climate Precipitation Center Morphing (CMORPH; Joyce et al., 2004) product to produce
an hourly 0.1° dataset from 2008. Starting from 2015, CMPA_1km upgrades CMPA by using more than 40000 automatic
weather stations and adding radar $P$ estimations in the merging procedure, which increases the spatial resolution to 1 km. Note
that precipitation data during the cold season (from October to April) in northern and western China are mainly derived from
remote sensing data since automatic weather stations do not operate under low temperatures (Shen et al., 2014; Shen et al.,
2018). SM data are obtained from the Soil Moisture Active Passive mission level 4 (SMAP-L4; Reichle et al., 2019) data.
SMAP-L4 assimilates SMAP radiometer brightness temperature into the NASA Catchment land surface model to produce a
3-hourly 9 km volumetric SM dataset in the root zone (0–100 cm). Since there is no direct measurement of SWE, we use snow
cover areas and snow depths as surrogates. Snow cover area (SCA) data are obtained from MOD10C1 (Hall and Higgs, 2021b)
and MYD10C1 (Hall and Higgs, 2021a), which collect snow extent information by the Moderate-Resolution Imaging



Spectroradiometer (MODIS) sensors from Terra and Aqua platforms respectively. Compiled at daily and 0.05° scales, MOD10C1 and MYD10C1 provide snow cover percentage and cloud cover percentage at each grid. Snow depth data are obtained from the long-term series of daily snow depth dataset in China (SD-CN; Che and Dai, 2015). The 25 km daily snow depth of SD-CN is derived from the passive microwave brightness temperature from SMMR, SSM/I, and SSMI/S sensors.

**Table 2. Sources of validation data.**

| Variable | Dataset | Spatial resolution | Temporal resolution | Temporal coverage | Reference |
|---|---|---|---|---|---|
| Precipitation | CMPA | 0.1° | hourly | 2008-2014 | Shen et al., 2014 |
| Precipitation | CMPA_1km | 1 km | hourly | 2015-2017 | Shen et al., 2018 |
| Soil moisture | SMAP-L4 | 9 km | 3-hourly | 2015-now | Reichle et al., 2019 |
| Snow cover area | MOD10C1 | 0.05° | daily | 2000-now | Hall and Higgs, 2021b |
| Snow cover area | MYD10C1 | 0.05° | daily | 2000-now | Hall and Higgs, 2021a |
| Snow depth | SD-CN | 25 km | daily | 1979-2019 | Che and Dai, 2015 |

## 3 Methods

The workflow of the study is presented in Fig. 1. Firstly, we reconstruct the precipitation data and then, we use the reconstructed precipitation as forcing input to reconstruct soil moisture and snow water equivalent.

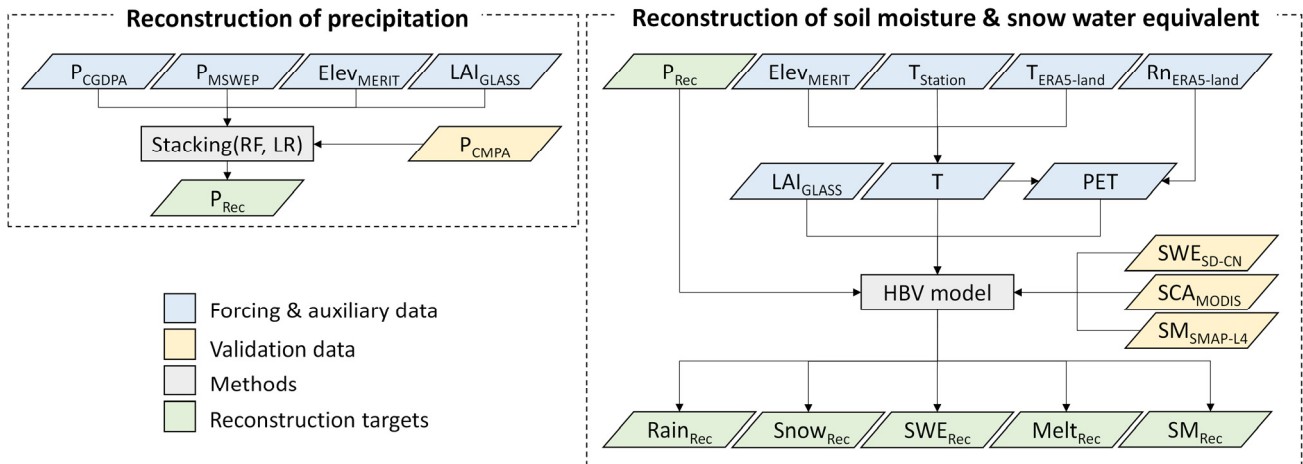

**Figure 1. Workflow of the study. P: precipitation, Elev: elevation, LAI: leaf area index, T: air temperature, Rn: net radiation, PET: potential evaporation, SWE: snow water equivalent, SCA: snow cover area, SM: soil moisture, Rain: liquid rainfall, Snow: snowfall, Melt: snowmelt, RF: random forest, LR: linear regression.**



### 3.1 Reconstruction of precipitation

We applied machine learning to predict CMPA precipitation grid by grid using grid coordinates, $P$ from CGDPA and MSWEP,
Elev, and LAI as input. All data were pre-processed to be daily and $0.1°$. MSWEP $P$ and CMPA $P$ were aggregated to be daily. GLASS LAI was set to be the same within the 8 days of an 8-day composite. The CMPA_1km $P$ and MERIT-Hydro Elev data were spatially aggregated into $0.1°$. The CGDPA $P$ data were resampled into $0.1°$ using bilinear interpolation. The difference between the ground-truth $0.1°$ $P$ and the resampled $0.1°$ $P$ from CGDPA is spatially correlated with the sub-grid distribution of Elev and LAI in a $0.25°$ grid. Therefore, we created two new input features, $Elev_{diff}$ and $LAI_{diff}$, to account for the differences
between the true $0.1°$ value and the value resampled from a $0.25°$ resolution for Elev and LAI. Specifically, Elev and LAI were aggregated into $0.25°$, resampled into $0.1°$ using bilinear interpolation, and then subtracted by the original $0.1°$ layer.

The training strategy is presented in Fig. 2. We divided the model training into two parts, a binary classification problem that predicted whether the grid was rainy ($P>0$ mm) and a regression problem which predicted the value of $P$ in rainy grids. We proposed a tile-by-tile training strategy that fit a machine learning model with all samples of one hundred $0.1°$ grids in a $1°$ tile.
To be specific, although the targeted $P_{CMPA}$ in a $0.1°$ grid was predicted by grid coordinates, $P_{CGDPA}$, $P_{MSWEP}$, Elev, $Elev_{diff}$, LAI, and $LAI_{diff}$ in the same grid, all grids in the same $1°$ tile shared a common prediction model. This tile-by-tile training strategy increased the size of samples and made use of the spatial information of Elev and LAI. Since the CMPA data were not reliable during the cold season in northern and western China because of lacking ground-based observations, we discarded training samples from October to April in two regions (Shen et al., 2018): (1) latitude>$40°$ N; (2) $40°$ N>latitude>$27°$ N and
longitude<$100°$ E.

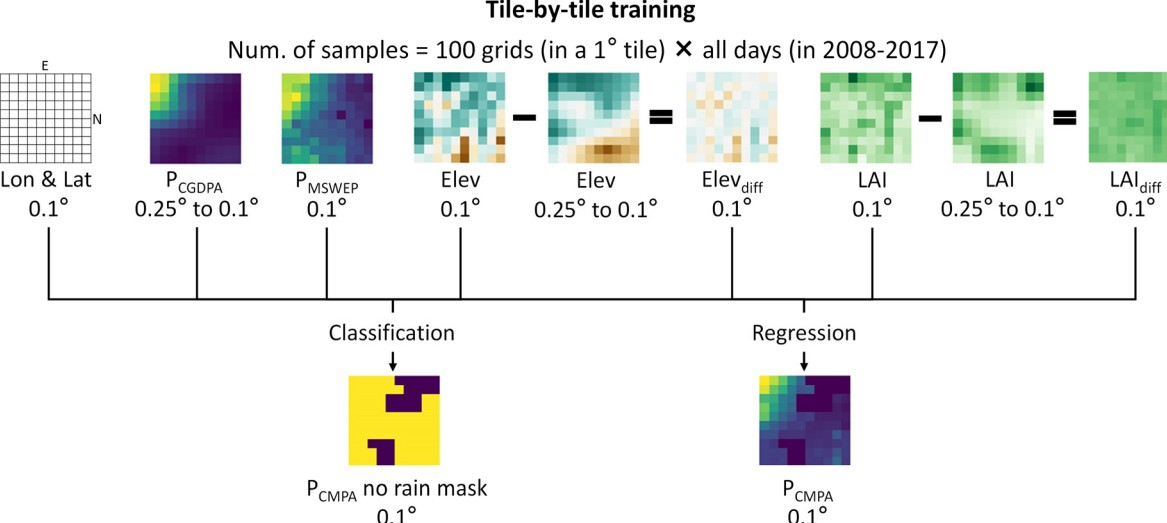


**Figure 2. Model training strategies of precipitation reconstruction. P: precipitation, Elev: elevation, LAI: leaf area index.**

The machine learning model for the rain/no rain classification is the random forest (Breiman, 2001). The model for the $P$ regression is a neural network (Foresee and Hagan, 1997) that stacks a linear regression model and a random forest model, as shown in Fig. 3. Stacking (Wolpert, 1992) is a model ensemble method that optimally combines multiple base machine learning models for predictions. In the stacking process, all base models are first trained to get out-of-bag predictions in the cross-validation, and then a stacking model is trained using the out-of-bag predictions of the base models as input. The random forest

model can deal with non-linear relationships and complex interactions among input features. The linear model can extrapolate the predictions that are out of the range of training samples. A stacking model leveraged the advantages of these two base models. We chose 5-fold cross-validation for hyper-parameter tuning and performance evaluation in both the classification and regression problems. After the model training, we used the data of $P_{\text{CGDPA}}$ and $P_{\text{MSWEP}}$ in 1981-2017 combined with grid coordinates, Elev, $\text{Elev}_{\text{diff}}$, LAI, and $\text{LAI}_{\text{diff}}$ to predict $P_{\text{CMPA}}$ in the same period, which produced a consistent long-term

reconstructed dataset $P_{\text{Rec}}$, as shown in Fig. 1. Note that we only predicted $P$ values on rainy days according to the classification model during the reconstruction.

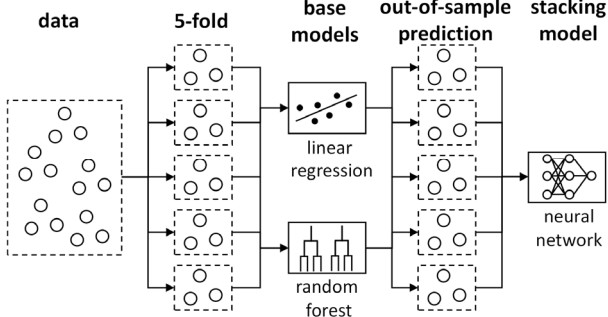

**Figure 3. Illustration of a stacking machine learning model for precipitation reconstruction.**


Validation metrics include two classification metrics, i.e., probability of detection (POD) and false alarm rate (FAR), and two regression metrics, i.e., Kling-Gupta Efficiency (KGE) and normalized root mean square error (NRMSE). Equations of the metrics are listed in Eq. 1 to 4:

$$\text{POD} = \frac{n_{11}}{n_{11}+n_{01}}, \tag{1}$$

$$\text{FAR} = \frac{n_{10}}{n_{11}+n_{10}}, \tag{2}$$

$$\text{KGE} = 1 - \sqrt{(r-1)^2 + \left(\frac{\mu_S}{\mu_O} - 1\right)^2 + \left(\frac{\sigma_S/\mu_S}{\sigma_O/\mu_O} - 1\right)^2}, \tag{3}$$


$$\mathrm{NRMSE} = \frac{\sqrt{\frac{1}{n}\sum_{i=1}^{n}(S_i - O_i)^2}}{\sigma_O}, \tag{4}$$

where $n_{11}$ is the number of actual rainy days that are predicted to be rainy, $n_{01}$ is the number of actual rainy days that are predicted to have no rain, $n_{10}$ is the opposite of $n_{01}$, $r$ is the correlation between predicted and observed $P$, $\mu_O$ and $\mu_S$ are the mean values of observed and predicted $P$ respectively, $\sigma_O$ and $\sigma_S$ are the standard deviations of observed and predicted $P$ respectively, $S_i$ is the predicted value of $P$ on the $i$th rainy day, $O_i$ is the observed value of $P$ on the $i$th rainy day. The perfect values are 1 for POD and KGE and 0 for FAR and NRMSE. Note that the predictions are validated in 5-fold cross-validation of the model. For grids in northern (latitude>40° N) and western (40° N>latitude>27° N and longitude<100° E) China, only predictions in May-September are validated.

## 3.2 Reconstruction of soil moisture and snow water equivalent

We used the modified HBV hydrological model (Bergström, 1992; Parajka et al., 2007) calibrated by satellite-based observations grid by grid to reconstruct SWE and SM. Note that we added a canopy interception module (Mao and Liu, 2019) to the traditional HBV model, as presented in Fig. 4. The forcing input of the model includes $P$, $T$, and potential evaporation (PET). The reconstructed $P$ was regarded as the precipitation input. The 0.1° $T$ was created by the interpolation of observations from ground stations using Co-Kriging (Myers, 1982) with Elev and the daily-aggregated ERA5-land $T$ as covariates. PET was calculated using Priestley-Taylor equation (Priestley and Taylor, 1972) with interpolated $T$ and daily-aggregated ERA5-land Rn.

The calibration targets were SCA, SWE, and SM, which should be pre-processed from the raw observational data before calibration. The 0.05° SCA data were first aggregated to be 0.1°. Then, for each grid, we extracted all days in November-April when the cloud cover percentages were smaller than 40% and calculated the percentage of snow extent in the no cloud fraction of the grid to get the final SCA. The calibration of SCA was converted to a binary classification problem where the observed SCA was set to be 1 if SCA>10% and 0 vice versa. In the HBV model, SCA was calculated by the binarization of SWE, which meant SCA=1 if SWE exceeded a snowpack threshold $T_{\mathrm{cover}}$ and SCA=0 vice versa. For SWE, we first resampled the 25 km snow depth into 0.1° using bilinear interpolation. Then, the observed SWE was calculated by multiplying snow depth and snow density. We used a fixed value of snow density at a national scale, 0.18 g cm$^{-3}$, which was reasonable for a range part of China (Yang et al., 2019; Gao et al., 2020; Yang et al., 2020). For SM, the 1 m root zone SM was resampled from 9 km to 0.1° using bilinear interpolation and aggregated from 3-hourly to daily.




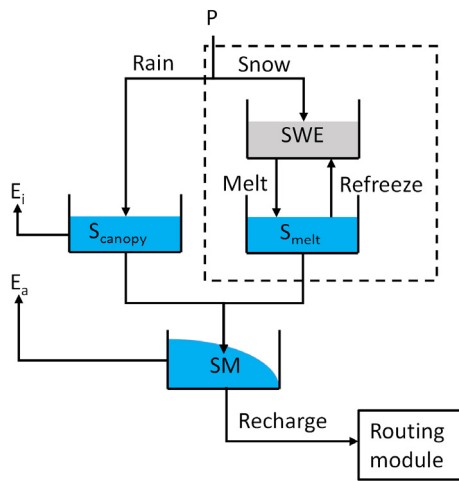


**Figure 4. Illustration of the HBV model used for the reconstruction of soil moisture and snow water equivalent. The dashed box illustrates the snow module.**

For each grid, we first calibrated the parameters of the snow module and then calibrated the soil water module with the optimal

snow module parameters. This two-step calibration strategy alleviated parameter equifinality since two modules were calibrated separately. The snow module was only calibrated in grids where at least 5% days with $T<0°C$. All parameters to be calibrated are presented in Table 3. After calibration, we used the historical $P_{Rec}$, PET, $T$, and LAI to drive the HBV model to simulate long-term rainfall ($Rain_{Rec}$), snowfall ($Snow_{Rec}$), snow water equivalent ($SWE_{Rec}$), snowmelt ($Melt_{Rec}$), and soil moisture ($SM_{Rec}$), as shown in Fig. 1.


**Table 3. Parameters of the HBV model and their ranges for model calibration.**

| Module | Parameters | Description | Range |
|---|---|---|---|
| Snow | TP | Critical temperature for rainfall and snowfall (°C) | [-3, 3] |
| | TM | Critical temperature for snowmelt and refreezing (°C) | [-3, 3] |
| | SCF | Correction factor for snowfall | [0.9, 1.5] |
| | CFX | Degree-day factor (mm °C$^{-1}$ d$^{-1}$) | [0.5, 10] |
| | CWH | Fraction of snowpack that can hold melt water | [0, 0.2] |
| | $T_{cover}$ | Snowpack threshold for 10% snow cover areas (mm) | [2, 10] |
| Soil | $m_c$ | Coefficient for interception storage capacity per unit LAI (mm) | [0.1, 0.5] |
| | FC | Soil storage capacity (mm) | [50, 800] |
| | beta | Shape coefficient in runoff generation curve | [0.1, 6] |
| | LP | Soil moisture above which soil evaporation reaches potential evaporation | [0.2, 1] |



The Validation metric for *SCA* is balanced accuracy (BACC):

$$\text{BACC} = \left( \frac{n_{11}}{n_{11}+n_{01}} + \frac{n_{00}}{n_{00}+n_{10}} \right)/2 \ , \tag{5}$$

where $n_{11}$ is the number of snow cover days that are predicted to have snow cover, $n_{00}$ is the number of no snow cover days that are predicted to have no snow cover, $n_{01}$ is the number of snow cover days that are predicted to have no snow cover, $n_{10}$ is the opposite of $n_{01}$. The validation metric for SWE and SM is KGE in Eq. 3, where $r$ is the correlation between predicted and observed SWE or SM, $\sigma_O$ and $\sigma_S$ are the standard deviations of observed and predicted SWE or SM respectively, $\mu_O$ and $\mu_S$ are the mean values of observed and predicted SWE or SM respectively. During the calibration, the optimization target of

the snow module was 1-1/4×BACC-3/4×KGE, which optimized the simulation performances of SCA and SWE at the same time. With small numbers of model parameters, the parsimonious snow and soil water modules were unlikely to overfit the observed data. Therefore, we validated the performance of the reconstruction using the performance of the calibration directly.

## 4 Results and discussion

### 4.1 Validation of precipitation

Table 4 summarises the performance of different $P$ datasets in all grids benchmarked by $P_{\text{CMPA}}$. For rain/no rain classification, the reconstructed $P$ ($P_{\text{Rec}}$) achieves a balance of the POD and FAR. The median POD of $P_{\text{Rec}}$ is 0.87, which is slightly worse than $P_{\text{CGDPA}}$ (POD=0.92) and far better than $P_{\text{MSWEP}}$ (POD=0.74). The median FAR of $P_{\text{Rec}}$ is 0.19, which is better than $P_{\text{CGDPA}}$ (FAR=0.23) but slightly worse than $P_{\text{MSWEP}}$ (FAR=0.18). Since the 0.1° $P_{\text{CGDPA}}$ is resampled from the 0.25° dataset, it certainly overestimates the probability of rain and has a high POD and FAR at the same time naturally. On the opposite, with the finer

spatial information from satellite data, $P_{\text{MSWEP}}$ is skilled in detecting no rain days and thus, it tends to have better FAR performance. In summary, $P_{\text{Rec}}$ improves the FAR of $P_{\text{CGDPA}}$ without scarifying too many POD. For $P$ regression, $P_{\text{Rec}}$ outperforms $P_{\text{CGDPA}}$ and $P_{\text{MSWEP}}$ significantly. The median KGEs of $P_{\text{Rec}}$, $P_{\text{CGDPA}}$, and $P_{\text{MSWEP}}$ are 0.68, 0.51, and 0.53 respectively and the median NRMSEs of $P_{\text{Rec}}$, $P_{\text{CGDPA}}$, and $P_{\text{MSWEP}}$ are 0.63, 0.91, and 0.82 respectively.

**Table 4. Validation of the reconstructed precipitation in all grids at the national scale. The benchmark dataset is CMPA. POD: probability of detection, FAR: false alarm rate, KGE: Kling-Gupta efficiency, NRMSE: normalized root mean squared error. In northern (latitude>40° N) and western (40° N>latitude>27° N and longitude<100° E) China, only data in May-September are used for validation.**

| Metric | Percentile | CGDPA | MSWEP | Reconstruction |
|---|---|---|---|---|
| | Min. | 0.18 | 0.03 | 0 |
| | 1st quartile | 0.85 | 0.62 | 0.81 |
| POD | Median | 0.92 | 0.74 | 0.87 |
| | 3rd quartile | 0.95 | 0.82 | 0.91 |
| | Max. | 1 | 0.96 | 1 |





| | | | | |
|---|---|---|---|---|
| | Min. | 0.01 | 0.01 | 0.01 |
| | 1st quartile | 0.17 | 0.12 | 0.14 |
| FAR | Median | 0.23 | 0.18 | 0.19 |
| | 3rd quartile | 0.3 | 0.24 | 0.25 |
| | Max. | 0.94 | 0.95 | 1 |
| | Min. | <-10 | <-10 | <-10 |
| | 1st quartile | 0.11 | 0.17 | 0.44 |
| KGE | Median | 0.51 | 0.53 | 0.68 |
| | 3rd quartile | 0.74 | 0.71 | 0.83 |
| | Max. | 0.96 | 0.93 | 0.97 |
| | Min. | 0.22 | 0.32 | 0.16 |
| | 1st quartile | 0.6 | 0.67 | 0.48 |
| NRMSE | Median | 0.91 | 0.82 | 0.63 |
| | 3rd quartile | 1.22 | 1.02 | 0.8 |
| | Max. | >10 | >10 | >10 |

Table 5 presents the median values of all metrics for $P_{Rec}$ in nine major basins of China. Figure 5 presents the spatial distribution of validation metrics for $P_{Rec}$ and the metric differences between $P_{Rec}$ and $P_{CGDPA}$ as well as $P_{Rec}$ and $P_{MSWEP}$. For rain/no rain classification, $P_{Rec}$ performs well since the median POD>0.83 and FAR<0.22 for all major basins except for the Southwest Basin, according to Table 5. According to Fig. 5 (a) and (d), the best performance occurs in the driest part of the Continental Basin where rain is rare, and the worst performance occurs in the plateau such as the upper stream areas of the Yangtze River

Basin and the Southwest Basin. The balance of POD and FAR can be seen in Fig. 5 (b), (c), (e), and (f), where $P_{Rec}$ trades POD for FAR compared with $P_{CGDPA}$ and trades FAR for POD compared with $P_{MSWEP}$. For $P$ regression, in addition to the Continental Basin and the upper stream of the Southwest Basin where the coverage of ground stations is low (Shen et al., 2014; Shen et al., 2018), all other major basins have median KGEs over 0.76 and NRMSEs under 0.55, which indicates good performance, according to Table 5. According to Fig. 5 (g) and (j), with KGE>0.8 and NRMSE<0.5 in a majority of grids,

$P_{Rec}$ is very accurate in a large part of the eastern region, probably because of the dense distribution of ground stations for CGDPA (Shen and Xiong, 2016). According to Fig 5 (h), (i), (k), and (l), $P_{Rec}$ outperforms $P_{CGDPA}$ and $P_{MSWEP}$ almost in the whole country. $P_{Rec}$ improves $P_{CGDPA}$ and $P_{MSWEP}$ the most in the Continental Basin and the upper stream of the Southwest Basin where the distribution of ground stations is sparse, even though the performance of $P_{Rec}$ is still limited in this area.

**Table 5. Validation of the reconstructed precipitation in nine major basins of China. The benchmark dataset is CMPA. POD: probability of detection, FAR: false alarm rate, KGE: Kling-Gupta efficiency, NRMSE: normalized root mean squared error. In northern (latitude>40° N) and western (40° N>latitude>27° N and longitude<100° E) China, only data in May-September are used for validation.**



| No. | Basin | Median POD | Median FAR | Median KGE | Median NRMSE |
|---|---|---|---|---|---|
| 1 | Continental Basin | 0.9 | 0.17 | 0.43 | 0.8 |
| 2 | Yangtze River Basin | 0.83 | 0.21 | 0.82 | 0.48 |
| 3 | Songhua and Liaohe River Basin | 0.84 | 0.22 | 0.76 | 0.55 |
| 4 | Southwest Basin | 0.75 | 0.27 | 0.43 | 0.8 |
| 5 | Yellow River Basin | 0.89 | 0.17 | 0.81 | 0.5 |
| 6 | Huaihe River Basin | 0.9 | 0.14 | 0.88 | 0.39 |
| 7 | Haihe River Basin | 0.92 | 0.13 | 0.86 | 0.42 |
| 8 | Southeast Basin | 0.86 | 0.17 | 0.88 | 0.39 |
| 9 | Pearl River Basin | 0.85 | 0.17 | 0.82 | 0.49 |






**Figure 5. Spatial validation of the reconstructed precipitation. The benchmark dataset is CMPA. POD: probability of detection, FAR: false alarm rate, KGE: Kling-Gupta efficiency, NRMSE: normalized root mean squared error. The second and third columns are the differences of validated metrics between the reconstructed dataset and the CGDPA and MSWEP datasets. The boundary lines delineate nine major river basins of China: 1. the Continental Basin, 2. the Yangtze River Basin, 3. the Songhua and Liaohe River Basin, 4. the Southwest Basin, 5. the Yellow River Basin, 6. the Huaihe River Basin, 7. the Haihe River Basin, 8. the Southeast Basin, 9. the Pearl River Basin. In northern (latitude>40° N) and western (40° N>latitude>27° N and longitude<100° E) China above the dashed lines, only data in May-September are validated.**





Figure 6 presents the spatial distribution of annual average $P$ and the time series of monthly $P$ for $P_{CMPA}$ and $P_{Rec}$. $P_{Rec}$ matches $P_{CMPA}$ well both spatially and temporally for nine major basins at a large temporal scale. According to Fig. 6 (a), $P_{Rec}$ does not smooth the spatial distribution of $P$, which indicates that the tile-by-tile training strategy learns the local variations of $P$ within the tile. Except for the Continental Basin and the Songhua and Liaohe River Basin where the cold season precipitation data are not reliable for CMPA, all other basins have KGE values larger than 0.91 for monthly time series, according to Fig. 6 (b).






**Figure 6. (a) Map of the annual average precipitation (P) of CMPA and reconstruction in 2008-2017. (b) Time series of the monthly P of CMPA and reconstruction for nine major river basins in 2008-2017. Note that CMPA has missing values in 2015, we only choose the days when CMPA has available data in the temporal aggregation for both CMPA and reconstruction. Here the reconstructed P**
**data are out-of-bag predictions in the cross-validation.**

## 4.2 Validation of soil moisture and snow water equivalent

Table 6 summarises the performance of reconstructed $SWE_{Rec}$, $SCA_{Rec}$, and $SM_{Rec}$ in all grids. The snow module of the HBV model performs poorly with the median KGE=-0.31 for $SWE_{Rec}$ at a national scale. While the snow module has certain skills
in simulating snow cover with the median BACC=0.65 for $SCA_{Rec}$. The soil water module of the HBV model performs well with the median KGE=0.61 for $SM_{Rec}$.

Table 7 presents the median values of all metrics for $SWE_{Rec}$, $SCA_{Rec}$, and $SM_{Rec}$ in nine major basins of China. Figure 7 presents the spatial distribution of validation metrics for $SWE_{Rec}$, $SCA_{Rec}$, and $SM_{Rec}$. The performance of snow reconstruction varies spatially. There are three major snow cover areas in China: northeast China, northern Xinjiang, and the Tibetan Plateau.
According to Fig. 7 (a) and (b), $SWE_{Rec}$ and $SCA_{Rec}$ perform well in both northeast China and northern Xinjiang with $KGE_{SWE}>0.7$ and $BACC_{SCA}>0.8$ in many grids. While $SWE_{Rec}$ and $SCA_{Rec}$ perform poorly with $KGE_{SWE}<0$ and $BACC_{SCA}<0.5$ in a large part of the Tibetan Plateau, where snow-driven hydrological processes are complex (Gao et al., 2020). According to Table 7, $SWE_{Rec}$ and $SCA_{Rec}$ perform the best in the Songhua and Liaohe River Basin (i.e., northeast China) with $KGE_{SWE}=0.55$ and $BACC_{SCA}=0.87$, where and snowmelt contributes a considerable amount of water to runoff and floods (Qi et al., 2021).
The performance of soil moisture reconstruction also varies spatially. According to Fig. 7 (c), $SM_{Rec}$ performs well in a large part of southern China. However, in the Continental Basin where the climate is dryer and the topography is more complex, $SM_{Rec}$ performs relatively poorly with median $KGE_{SM}=0.3$. Generally, $SM_{Rec}$ performs better in southern basins, e.g., the Yangtze River Basin, the Huaihe River Basin, the Southeast Basin, and the Pearl River Basin, where the values of median $KGE_{SM}$ are at least 0.76, according to Table 7. Note that the accuracy of $SWE_{Rec}$, $SCA_{Rec}$, and $SM_{Rec}$ in different areas may
depend on the quality of the benchmark datasets and the ability of the HBV model to represent local hydrological processes.

**Table 6. Validation of the reconstructed snow water equivalent (SWE), snow cover area (SCA), and soil moisture (SM) in all grids at the national scale. The benchmark datasets are SD-CN for SWE, MOD10C1/MYD10C1 for SCA, and SMAP-L4 for SM. KGE: Kling-Gupta efficiency, BACC: balanced accuracy.**

| Percentile | $KGE_{SWE}$ | $BACC_{SCA}$ | $KGE_{SM}$ |
|---|---|---|---|
| Min. | <-10 | 0.22 | -1.83 |
| 1st quartile | -4.76 | 0.54 | 0.33 |
| Median | -0.31 | 0.65 | 0.61 |
| 3rd quartile | 0.36 | 0.82 | 0.80 |
| Max. | 0.90 | 1.00 | 0.99 |






**Table 7. Validation of the reconstructed snow water equivalent (SWE), snow cover area (SCA) in nine major basins of China. The benchmark datasets are SD-CN for SWE, MOD10C1/MYD10C1 for SCA, and SMAP-L4 for SM. KGE: Kling-Gupta efficiency, BACC: balanced accuracy.**

| No. | Basin | Median KGE$_{SWE}$ | Median BACC$_{SCA}$ | Median KGE$_{SM}$ |
|---|---|---|---|---|
| 1 | Continental Basin | -2.41 | 0.64 | 0.3 |
| 2 | Yangtze River Basin | -0.57 | 0.53 | 0.79 |
| 3 | Songhua and Liaohe River Basin | 0.55 | 0.87 | 0.7 |
| 4 | Southwest Basin | -0.85 | 0.6 | 0.59 |
| 5 | Yellow River Basin | 0.16 | 0.57 | 0.65 |
| 6 | Huaihe River Basin | 0.3 | 0.54 | 0.76 |
| 7 | Haihe River Basin | 0.37 | 0.64 | 0.73 |
| 8 | Southeast Basin | — | — | 0.8 |
| 9 | Pearl River Basin | — | — | 0.88 |

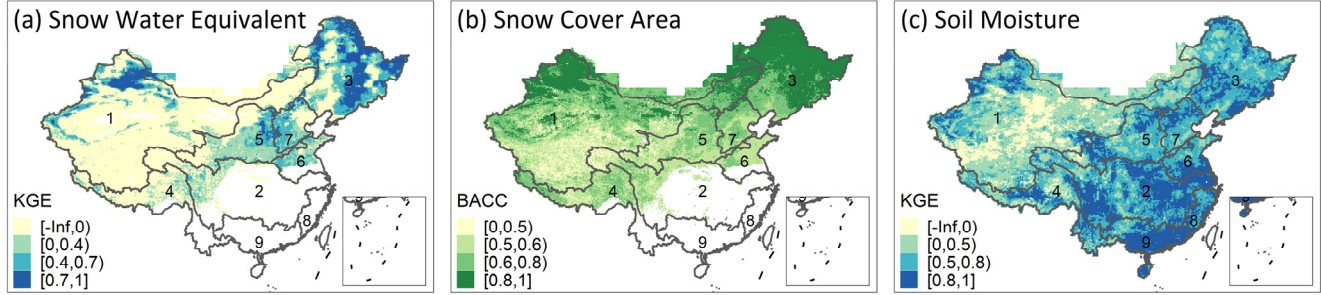

**Figure 7. Spatial validation of the reconstructed snow water equivalent (SWE), snow cover area (SCA), and soil moisture (SM). The benchmark datasets are SD-CN for SWE, MOD10C1/MYD10C1 for SCA, and SMAP-L4 for SM. KGE: Kling-Gupta efficiency, BACC: balanced accuracy. The boundary lines delineate nine major river basins of China: 1. the Continental Basin, 2. the Yangtze River Basin, 3. the Songhua and Liaohe River Basin, 4. the Southwest Basin, 5. the Yellow River Basin, 6. the Huaihe River Basin, 7. the Haihe River Basin, 8. the Southeast Basin, 9. the Pearl River Basin.**

Figure 8 presents the spatial distribution of daily average SWE and the time series of daily average SWE for SWE$_{SD-CN}$ and SWE$_{Rec}$ in each month. Fig. 8 (a) shows that SWE$_{Rec}$ successfully detects areas with large SWE. Fig. 8 (b) shows that, although SWE$_{Rec}$ captures the temporal variations of SWE in all basins, it overestimates the magnitudes of SWE in the Continental Basin, the Yangtze River Basin, the Southwest Basin, and the Yellow River Basin. In the Songhua and Liaohe River Basin with KGE=0.68 and the Haihe River Basin with KGE=0.46, SWE$_{Rec}$ can accurately capture both the temporal variations and the magnitudes of SWE. The magnitudes of SWE are difficult to simulate for three reasons. First, SWE$_{SD-CN}$ is not the actual observed SWE but an estimation of SWE from the multiplication of snow depth and a fixed snow density; second, the original spatial resolution of SWE$_{SD-CN}$ is 25km, which may be too coarse to represent snow distribution, especially in mountain regions;





third, the new reconstructed precipitation dataset may not capture snowfall well. Figure 9 presents the spatial distribution of daily average SM and the time series of daily average SM for $SM_{SMAP-L4}$ and $SM_{Rec}$ in each month. $SM_{Rec}$ captures the spatial and temporal variations of $SM_{SMAP-L4}$ well except in the Continental Basin. Although the monthly KGE values in all basins are larger than 0.59 except for the Continental Basin, $SM_{Rec}$ slightly underestimates $SM_{SMAP-L4}$ at the monthly scale in all basins.

**Figure 8. (a) Map of daily average snow water equivalent (SWE) of SD-CN and reconstruction in 2000-2017. (b) Time series of the daily average SWE of SD-CN and reconstruction for nine major river basins in each month of 2000-2017.**








**Figure 9. (a) Map of daily average soil moisture (SM) of SMAP-L4 and reconstruction in 2015-2017. (b) Time series of the daily average SM of SMAP-L4 and reconstruction for nine major river basins in each month of 2015-2017.**

### 4.3 Limitations of the reconstruction dataset

Uncertainties of the reconstruction dataset come from the quality of the input data and the limitations of the reconstruction models. For precipitation, although the benchmarked dataset CMPA includes satellite information to produce $P$ data, it still



has large errors in western China due to the small number of automatic weather stations for local corrections in this area. In addition, the quality of MSWEP is not consistent over time since the available data sources to be merged are changing in different periods. Furthermore, although the stacking machine learning model can extrapolate, it may have problems in reconstructing extreme $P$ values since the extrapolation relies on a linear regression model, which may fail to capture the

complex relationship between the targeted $P$ and input variables. For snow water equivalent, a unified snow density (0.18 g cm$^{-3}$) may cause a large bias in estimating SWE in some regions. The coarse resolution of the benchmarked SWE is also a major concern: although the MODIS SCA data add sub-grid information to the 0.25° SD-CN data, we still do not have a benchmarked SWE dataset that is originally 0.1°. Another limitation is the unknown applicability of the HBV model at a national scale. HBV uses a temperature-based snow module without an energy balance component or a glacier module, which

may fail in areas with more complex snow processes such as the Tibetan Plateau (Gao et al., 2020). For soil moisture, the benchmarked SMAP-L4 is an assimilated dataset without actual measurements of root zone SM. In addition, PET is calculated without local $R_n$ data. The uncertainty of PET may propagate to SM. Moreover, it is unclear whether HBV is suitable for soil water simulation at a national scale in China.

**5 Conclusions**

We created a long-term (1981-2017) 0.1° daily dataset of total precipitation ($P$), liquid rainfall (Rain), snowfall (Snow), snow water equivalent (SWE), snowmelt (Melt), and soil moisture (SM) in China by reconstructing high-resolution satellite-based data. $P$ was reconstructed by predicting CMPA data from CGDPA and MSWEP data using a stacking machine learning model. Other variables were simulated by the HBV model with SWE calibrated by SD-CN, SCA calibrated by MOD10C1/MYD10C1, and SM calibrated by SMAP-L4. Evaluations of the reconstruction data are as follows.

- Benchmarked by CMPA at a national scale, the median POD and FAR of the reconstructed $P$ are 0.87 and 0.19 respectively for rain/no rain classification, and the median KGE and NRMSE of the reconstructed $P$ are 0.68 and 0.63 respectively for $P$ regression in rainy days. The reconstructed $P$ improves the CGDPA and MSWEP data, whose median KGEs are 0.51 and 0.53. The median KGEs are smaller than 0.43 in the Continental Basin and the Southwest Basin and larger than 0.76 in other major basins. At the monthly scale, all basins have KGE values larger than 0.91 except for the Continental Basin

and the Songhua and Liaohe River Basin, where the benchmarked precipitation data in cold seasons are not reliable.
   - Benchmarked by SD-CN and MOD10C1/MYD10C1 at a national scale, the median KGE of the reconstructed SWE and the median BACC of the reconstructed SCA are -0.31 and 0.65 respectively. The reconstructed SWE performs the best in the Songhua and Liaohe Basin with KGE=0.55 but performs the worst in the Continental Basin where the median KGE is -2.41. At the monthly scale, the reconstructed SWE captures the monthly variability of observed SWE in all basins.

However, the reconstructed SWE only reproduces SWE magnitudes accurately in the Songhua and Liaohe Basin (monthly KGE=0.68) and the Haihe Basin (monthly KGE=0.46) but overestimates the SWE magnitudes in other basins.

- Benchmarked by SMAP-L4 at a national scale, the median KGE of the reconstructed SM is 0.61. $SM_{Rec}$ performs well in southern basins, e.g., the Yangtze River Basin, the Huaihe River Basin, the Southeast Basin, and the Pearl River Basin, where the values of median $KGE_{SM}$ are at least 0.76. $SM_{Rec}$ performs the worst in the Continental Basin where the median

$KGE_{SM}$=0.3. At the monthly scale, the KGE values in all basins are larger than 0.59 except for the Continental Basin.

This study is the first attempt to produce a long-term (at least 30 years) 0.1° daily dataset of $P$, SM, and SWE that combines high-accuracy local information and high-resolution satellite-based data via reconstruction. This dataset is especially suitable for exploring the relationship between riverine streamflow and hydrological drivers since the $P$, SM, and SWE are produced independently from streamflow data. Future improvements include extending the temporal length of the dataset and

formulating a model strategy that handles the spatial variability of the hydrological processes at a national scale.

**Code availability**

The source codes for model training, model calibration, and data reconstruction are available at https://github.com/YANGOnion/Hydrological-Reconstruction-China.

**Data availability**

The reconstruction datasets of total precipitation, rainfall, snowfall, snow water equivalent, snowmelt, and soil moisture are freely available at https://doi.org/10.5281/zenodo.5811099 (Yang et al., 2021).

**Acknowledgements**

We thank the China Meteorological Administration for providing national precipitation data (CGDPA, CMPA, and
CMPA_1km) and air temperature data from meteorological stations.

**Author contribution**

WY and HY conceptualized the study. WY developed the methodology, performed the analysis and wrote the original draft of manuscript. All other authors contributed to review and revise the manuscript. HY, QH, and DY helped in the data collection. HY and DY were responsible for funding acquisition.

**Competing interests**

The authors declare that they have no conflict of interest.



**Financial support**

This research was supported by the China National Key R&D Program (grant no. 2021YFC3000202), the National Natural Science Foundation of China (grant no. 51979140, 42041004) and the State Key Laboratory of Hydro Science and Hydraulic Engineering of China (grant no. 2021-KY-04) .

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
