# Peer review of "Long-term reconstruction of satellite-based precipitation, soil moisture, and snow water equivalent in China"

_Hydrology and Earth System Sciences, 2022_

## Author Comment (AC2)

**Reviewer #2 Comment 1:** (hereafter referred to as R2C1, R2C2…) *This manuscript reconstructed a long-term precipitation data set using a stacking machine learning model according to MSWEP and CGDPA, and at the same time reconstructed long-term soil moisture and snow water equivalent data sets using the HBV model. The reconstructed data ranges from 1981-2017 and with a high spatial-temporal resolution, i.e. a 0.1 degree spatial resolution and daily temporal resolution. The results show that the data has a high reliability, especially the precipitation data. It provides valuable products across China for meteorological and hydrological domains, such hydrological modeling, assessing the hydrological response under climate change and etc. The manuscript is well-organized and well-written. It is suitable for Hydrology and Earth System Sciences. Therefore, I recommend a minor revision.*

**A:** Thank you for your constructive comments. We have carefully considered your suggestions for the revision of the manuscript.

*Detailed comments:*

**R2C2:** *1. The spatial data used in this manuscript has different units, namely degree and km. Especially, the variables with different units, including precipitation, soil moisture, snow cover, and snow depth, were used for calibrating and validating HBV model. It isn't clear which geographic and projected coordinate systems were chosen and which unit were used for calibrating and validating.*

**A:** Thank you very much for your critical comment. The HBV model is calibrated and validated at a 0.1° resolution under the WGS 84 latitude/longitude coordinate system (EPSG:4326). Therefore, the forcing, auxiliary, and validation data for HBV were pre-processed to be 0.1° under EPSG:4326. We mentioned the way to unify the spatial resolutions of different datasets in the original manuscript. For example, "The calibration targets were SCA, SWE, and SM, which should be pre-processed from the raw observational data before calibration. The 0.05° SCA data were first aggregated to be 0.1°." (Line94-95); "For SWE, we first resampled the 25 km snow depth into 0.1° using bilinear interpolation." (99-100); "For SM, the 1 m root zone SM was resampled from 9 km to 0.1° using bilinear interpolation and aggregated from 3-hourly to daily." (Line 103-104). In the revised manuscript, we will specify the coordinate system we use and clarify the requirement of spatial resampling for different datasets before hydrological modeling.

**R2C3:** *2. Two types of precipitation, namely CMPA from 2008 to 2014 and CMPA_1km from 2015 to 2017, were used for machine learning model. The uncertainty due to the precipitation inconsistency should be discussed.*

**A:** Thank you for your important comment. Yes, the inconsistency of CMPA data over time may raise uncertainties. As the successor of CMPA, CMPA_1km retains most of the observation sources that are used in CMPA, including more than 30000 automatic weather stations and the CMORPH product. In addition, The main deriving methods of CMPA and CMPA_1km are similar, including optimal interpolation and local bias correction. Therefore, the difference between CMPA and CMPA_1km is not significant in theory. Considering that data sizes contribute a lot to the accuracy of machine learning predictions, it is reasonable to combine CMPA and CMPA_1k which can trade consistency for more data samples.

**R2C4:** *3. The abstract stated "the short-term 0.1o CMPA", and I suggest clarifying time frame on the short-term.*

**A:** Thank you very much for your valuable comment. The "short-term" phrase refers to the period of CMPA (2008-2017) compared with the long period of the reconstruction data (1981-2017). We will specify the period instead of using "short-term" in the revised manuscript.

**R2C5:** *4. On the names of the nine major river basins of China, I suggest replacing the Southwest Basin with the Southwest Basins, and replacing the Southeast Basin with the Southeast Basin, since either of them includes more than one basin.*

**A:** Thank you very much for your comments. A large basin consists of many sub-basins. The names "Southwest Basin" and "Southeast Basin" are used in the official boundary data of nine major basins in China (https://www.resdc.cn/data.aspx?DATAID=141). Therefore, we prefer to keep these names.

---

## Author Response (AR1)

Dear Editor and Reviewers,

Thank you very much for the assessment of our manuscript. We appreciate that you provide an opportunity for us to improve our study. We have revised our manuscript thoroughly according to the comments. Please see the replies to the reviewers' comments below. In the response, the blue texts are the comments and the green texts are the quotes of the manuscript.

On behalf of all co-authors,
Wencong Yang

**Reviewer #1 Comment 1:** (hereafter referred to as R1C1, R1C2…) *Based on data fusion and numerical modeling, the authors reconstructed a long-term (1981-2017) 0.1° daily dataset of total precipitation (P), liquid rainfall (Rain), snowfall (Snow), snow water equivalent (SWE), snowmelt (Melt), and soil moisture (SM) in China. Reconstructing these hydrological components is a very challenging topic, particularly for west China including the Northwest and Tibetan Plateau. The reader may expect some progress in these regions when seeing the title, but the results of this study show limited progress in these regions. This is certainly not surprising, and it looks like there is a long way to go. Nevertheless, the authors took advantage of two favorable conditions to extend the recent data with better accuracy to the 1980s, which is a clear progress. That is, (1) meteorological data and satellite data in the last decade or so are more abundant, higher resolution and more accurate; (2) Satellite remote sensing data can be used to calibrate a land hydrological model. They are the innovative points of this paper and support its publication.*

**A:** Thank you for your constructive comments. We have carefully considered your suggestions for the revision of the manuscript. We would like to reaffirm the contribution of this study. Recent state-of-the-art satellite-based data have high spatial resolutions and abundant observational information with full spatial coverage. The objective of this study is to extend the satellite-based data of *P*, SWE, and SM to the 1980s in China so that hydrological studies focusing on a longer time span can take advantage of those high-quality data. The innovations you summarized match the contribution of this study except for one point, i.e., we do not aim to create a dataset with "better accuracy" compared with the satellite-based data used for model training or calibration. The satellite-based data are the benchmarks of the reconstruction dataset

we produced. So the accuracy of the reconstruction dataset is not able to exceed that of the satellite-based data. In the revised manuscript, we added a paragraph about the value and limitation of reconstruction data in Section 4.3 as "Reconstruction data do not have better accuracy than the satellite-based benchmark data we used. Instead, reconstruction aims to extend the state-of-the-art satellite-based data of P, SM, and SWE to the 1980s in China. Therefore, the value of the reconstruction data is to support hydrological studies focusing on a longer time span (e.g., 30 years) rather than recent years." (Line 378-380)

According to the objective of the study we pinpoint above, we do make some progress in the Northwest and Tibetan Plateau but not in the aspect you expected. Improvements are made in those regions not for the latest years but for the historical period when multiple-sources satellite-based observations are not available. It is challenging to comprehensively evaluate any new dataset in the Northwest and Tibetan Plateau since the "ground truth" information, i.e., observations from ground-based stations, are very limited in those regions.

*My main comments are as follows.*

**R1C2:** *Research focus. From the results, the reconstructed P data are of high quality and may be used by other researchers. However, the significance of the snow data and soil moisture content data is weaker. Especially for snow data, this paper uses the data of Che et al. as training data and the evaluation is also based on the data and there is no validation based on independent data. Given that the time series of Che et al. data is even longer (1979-2020), the snow data presented in this study is not very necessary. I suggest that the snow-related section be greatly weakened, and its validation section could even be removed.*

**A:** Thank you very much for your critical comment. We agree with you that the significance of the reconstructed *P*, SWE and SM data varies based on the quality. However, the reconstructed SWE and SM data have their unique values. As we stated in the introduction section, existing long-term high-resolution SWE and SM data in China are the byproduct of runoff modeling without direct calibrations on SWE and SM. Therefore, this study takes a step further to calibrate hydrological models with satellite-based data so that the reconstruction data become the historical extension of the recent satellite-based data. The reconstructed SM data are accurate with the median KGE=0.61 across the country at the daily scale (Table 6). Although the reconstructed SWE data fail to match the satellite-based snow data in the Tibetan Plateau, they are

relatively accurate in the Songhua and Liaohe River Basin with median KGE=0.55 across the basin at the daily scale (Table 7). In addition, the reconstructed SWE data have certain skills in matching the satellite-based SWE in northern Xinjiang (Figure 7 a and b) which is another snow hydrology hotspot region.

The SD-CN (Che and Dai, 2015) dataset is indeed long enough to cover the reconstruction period of this study. However, SD-CN has a coarse spatial resolution (25km), which does not satisfy the 0.1° resolution we intend to reach. Therefore, we prefer to keep the validation section of snow data. In the revised manuscript, we added a comment about SD-CN in Line 128-129 as "Although SD-CN is long enough for many hydrological studies, it is limited by a coarse spatial resolution."

**R1C3:** *The abstract needs to describe that (1) the constructed P is used to drive HBV, which generates snow and SM data, and (2) the reconstruction algorithm and data evaluation use the same data sources, only the data periods are different (if my understanding is correct). The current description will make the reader struggle to find the difference between the training data and validation data.*

**A:** Thank you for your important comment. (1) We added a sentence "The reconstructed P data are used to drive HBV hydrological model to simulate SM and SWE data in 1981-2017." in the abstract (Line 18-19). (2) Yes, for reconstructed *P* data, the reconstruction algorithm and data evaluation use the same data sources but with different periods. We changed the description of evaluation to "Cross-validated by the spatial and temporal splitting of CMPA data, the median Kling-Gupta Efficiency (KGE) of the reconstructed *P* is 0.68 for all grids at a daily scale." (Line 21-22).

**R1C4:** *Data description. (1) The use of CMPA_1km is not clearly stated. The authors just say upscale to 0.1 degree, without stating how the 1km-resolution data is used. (2) It should be made clear that the data in Table 2 are both for training and for validation. If this is not made clear, it is difficult to understand the structure of Figure 1. This once made the reviewer confused. (3) The data introduction section suddenly mentions air temperature and net radiation without introducing the usage of these data. This may confuse the reader.*

**A:** Thank you very much for your valuable comment. (1) The CMPA_1km data in 2015-2017 were upscaled to 0.1 degrees to act as the temporal extension of the original CMPA data in 2008-2014. We added the use of CMPA_1km in Line 115-117 as "In this study, CMPA_1km was spatially aggregated into 0.1° to extend the time span of

the CMPA data. CMPA refers to the combination of CMPA and CMPA_1km in the following part of the paper." (2) In the revised manuscript, we clarified the use of data in Table 2 as "The validation data are the reconstruction targets of the study, and therefore, they are also used for model training (for precipitation) and calibration (for soil moisture and snow water equivalent). Details about validation methods are introduced in Section 3." (Line 108-110). (3) The temperature and net radiation are used to drive HBV model. As we stated in the method section in Line 206-208 "PET was calculated using Priestley-Taylor equation (Priestley and Taylor, 1972) with interpolated T and daily-aggregated ERA5-land Rn." In the revised manuscript, we added the use of temperature and net radiation data in the data description section as "In addition to precipitation data, hydrological modeling requires air temperature (T) and net radiation data (Rn) as forcing data." (Line 98-99)

**R1C5:** *Methodological aspects. (1) It is not clear why the HBV model is used, and what are the advantages in reflecting SM and SWE. (2) It needs to be clarified what exactly is meant by 5-fold cross validation, e.g. whether this fold is temporal or spatial, which time periods (or spaces) are used for training and which time periods (or spaces) for validation. This information must be clearly, perhaps in Table 2.*

**A:** Thank you very much for your comments. (1) The HBV model is used because of its low computational complexity and general applicability in various climate conditions (Beck et al., 2020; Seibert and Bergström, 2022). In addition, HBV models show high capability in simulating soil moisture in various regions of the world (Beck et al., 2021). In the revised manuscript, we added the advantages of HBV in Line 200-202 as "The HBV model has low computational complexity and general applicability in various climate conditions (Beck et al., 2020; Seibert and Bergström, 2022) and show high capability in simulating soil moisture in various regions of the world (Beck et al., 2021)." (2) We performed a spatial and temporal mixed 5-fold cross-validation, i.e., we put the data of one hundred $0.1°$ grids in a $1°$ tile on all days together to form a training dataset and then split it into 5 folds randomly. Each sample corresponds to the data of one $0.1°$ grid on one day. In the revised manuscript, we clarified the strategy of cross-validation in Line 173-175 as "All folds were created by the spatial and temporal mixed splitting of the data samples. Specifically, we put the data of one hundred $0.1°$ grids in a $1°$ tile on all days together to form a training dataset and then split it into 5 folds randomly. Each sample corresponded to the data of one $0.1°$ grid on one day."

Beck, H. E., Pan, M., Lin, P., Seibert, J., van Dijk, A. I., & Wood, E. F.: Global fully distributed parameter regionalization based on observed streamflow from 4,229 headwater catchments, J Geophys Res-Atmos, 125(17), 10.1029/2019jd031485, 2020.

Beck, H. E., Pan, M., Miralles, D. G., Reichle, R. H., Dorigo, W. A., Hahn, S., ... & Wood, E. F.: Evaluation of 18 satellite-and model-based soil moisture products using in situ measurements from 826 sensors, Hydrol. Earth Syst. Sci., 25(1), 17-40, 10.5194/hess-25-17-2021, 2021.

Seibert, J., and Bergström, S.: A retrospective on hydrological catchment modelling based on half a century with the HBV model, Hydrol. Earth Syst. Sci., 26(5), 1371-1388, 10.5194/hess-26-1371-2022, 2022.

**R1C6:** *Validation issues. (1) The data sources used in the current evaluation are the same as those used for training, but validations based on independent data are more convincing. For example, precipitation data need to be validated at stations of pre-CMPA era and SM data need to be validated based on intensive SM observation networks (e.g., the widely used Maqu and Naqu soil moisture measuring networks). (2) Figure 6 shows that overall the reconstruction of P is good, but why are its errors significantly larger in several watersheds in 2017?.*

**A:** Thank you very much for your comment. (1) On the one hand, we adopt 5-fold cross-validation (see R1C5) to train and validate reconstruction models for precipitation. It means that the validation is independent from the training data. On the other hand, as we stated in R1C1, we do not aim to create a dataset with the best accuracy compared with all existing data products. Instead, we intend to extend the state-of-the-art but short-term data of *P*, SWE, and SM to the 1980s. Therefore, validations should present the similarity between the reconstruction data and the target data. Validations based on independent station-based data are out of the scope of this study. (2) In fact, a small discrepancy between reconstructed *P* and CMPA data occur since 2015 according to Figure 6. The reason is that the CMPA data used in this study are not perfectly consistent across time. As introduced in section 2.2, the CMPA data after 2015 come from the upscaling CMPA_1km data. Compared with CMPA, CMPA_1km has additional observational data sources including more automatic weather stations and radar estimations. Therefore, system bias may occur between CMPA and CMPA_1km. However, combining CMPA and CMPA_1k increases data sizes, which contributes a lot to the accuracy of machine learning predictions. In the revised manuscript, we

regarded the inconsistency between CMPA and CMPA_1km as one of the uncertainty sources of the reconstruction data and added a statement in Line 363-366 as "Another problem of CMPA data is the possible inconsistency between CMPA and its successor, CMPA_1km. Although the main observation sources and the deriving method are similar, the discrepancy between CMPA and CMPA_1km has not been investigated in previous studies. Combining CMPA and CMPA_1km trades consistency for more data samples in machine learning algorithms."

*Other comments:*

**R1C7:** *The terms "ground-truth" or "raw observational data" or "observed SWE" are mentioned in the text, but please avoid using them in this way, because in fact they refer to fused data or remotely sensed data.*

**A:** Thank you very much. We have changed the description to "target data" or "target SWE" accordingly in the revised manuscript.

**R1C8:** *It is difficult to understand the Continental Basin, suggest to change to NW Continental Basin*

**A:** Thank you very much for your comment. The name "Continental Basin", also known as the "Inland River Basin", is often used in hydrological research. In addition, the name "Continental Basin" is used in the official boundary data of nine major basins in China (https://www.resdc.cn/data.aspx?DATAID=141). Therefore, we prefer to keep this name.

**R1C9:** *P3: "For P, we merged CGDPA and MSWEP to reconstruct the P from CMPA using machine learning techniques; for SM, we used the reconstructed P to drive a hydrological model to reconstruct SM from SMAP level 4". This description (reconstruct the P from CMPA, reconstruct SM from SMAP) is quite confusing.*

**A:** Thank you very much. In the revised manuscript, the sentence was revised as "We merged CGDPA and MSWEP to reconstruct the *P* benchmarked by CMPA using machine learning techniques. We used the reconstructed P to drive a hydrological model to reconstruct SM calibrated by SMAP level 4." (Line 82-84)

**R1C10:** *P8: "For SM, the 1 m root zone SM …". Although I know what it refers to, for most readers it may not be clear that it is SMAP-L4.*

**A:** Thank you very much. The sentence has been changed to "For SM, the 1 m root zone SM from SMAP-L4 was resampled from 9 km to 0.1° using bilinear interpolation and aggregated from 3 hours to 1 day." (Line 217-218)

**R1C11:** *P10: Start a new paragraph from "The validation metric for SWE and SM is KGE in Eq. 3".*

**A:** Thank you very much. We have modified the paragraph in the revised manuscript. (Line 238)

**Reviewer #2 Comment 1:** (hereafter referred to as R2C1, R2C2…) *This manuscript reconstructed a long-term precipitation data set using a stacking machine learning model according to MSWEP and CGDPA, and at the same time reconstructed long-term soil moisture and snow water equivalent data sets using the HBV model. The reconstructed data ranges from 1981-2017 and with a high spatial-temporal resolution, i.e. a 0.1 degree spatial resolution and daily temporal resolution. The results show that the data has a high reliability, especially the precipitation data. It provides valuable products across China for meteorological and hydrological domains, such hydrological modeling, assessing the hydrological response under climate change and etc. The manuscript is well-organized and well-written. It is suitable for Hydrology and Earth System Sciences. Therefore, I recommend a minor revision.*

**A:** Thank you for your constructive comments. We have carefully considered your suggestions for the revision of the manuscript.

*Detailed comments:*

**R2C2:** *1. The spatial data used in this manuscript has different units, namely degree and km. Especially, the variables with different units, including precipitation, soil moisture, snow cover, and snow depth, were used for calibrating and validating HBV model. It isn't clear which geographic and projected coordinate systems were chosen and which unit were used for calibrating and validating.*

**A:** Thank you very much for your critical comment. The HBV model is calibrated and validated at a 0.1° resolution under the WGS 84 latitude/longitude coordinate system (EPSG:4326). Therefore, the forcing, auxiliary, and validation data for HBV were pre-processed to be 0.1° under EPSG:4326. We mentioned the way to unify the spatial resolutions of different datasets in the original manuscript. For example, "The calibration targets SCA, SWE, and SM were pre-processed from the raw data in Table 2. The 0.05° SCA data from MOD10C1 or MYD10C1 were first aggregated to be 0.1°." (Line 209-210); "For SWE, we first resampled the 25 km snow depth from SD-CN into 0.1° using bilinear interpolation." (214-215); "For SM, the 1 m root zone SM from SMAP-L4 was resampled from 9 km to 0.1° using bilinear interpolation and aggregated from 3 hours to 1 day." (Line 217-218). In the revised manuscript, we specified the coordinate system we used for the precipitation reconstruction method (Line 144-145, "All data were pre-processed to be daily and 0.1° under the WGS 84 latitude/longitude coordinate system (EPSG:4326).") and for the HBV model (Line 203-204, "The HBV

model is calibrated at a 0.1° resolution under the WGS 84 latitude/longitude coordinate system (EPSG:4326).").

**R2C3:** *2. Two types of precipitation, namely CMPA from 2008 to 2014 and CMPA_1km from 2015 to 2017, were used for machine learning model. The uncertainty due to the precipitation inconsistency should be discussed.*

**A:** Thank you for your important comment. Yes, the inconsistency of CMPA data over time may raise uncertainties. As the successor of CMPA, CMPA_1km retains most of the observation sources that are used in CMPA, including more than 30000 automatic weather stations and the CMORPH product. In addition, The main deriving methods of CMPA and CMPA_1km are similar, including optimal interpolation and local bias correction. Therefore, the difference between CMPA and CMPA_1km is not significant in theory. Considering that data sizes contribute a lot to the accuracy of machine learning predictions, it is reasonable to combine CMPA and CMPA_1k which can trade consistency for more data samples. In, we added the statement about the uncertainty above in Section 4.3 as "Reconstruction data do not have better accuracy than the satellite-based benchmark data we used. Instead, reconstruction aims to extend the state-of-the-art satellite-based data of P, SM, and SWE to the 1980s in China. Therefore, the value of the reconstruction data is to support hydrological studies focusing on a longer time span (e.g., 30 years) rather than recent years." (Line 378-380)

**R2C4:** *3. The abstract stated "the short-term 0.1o CMPA", and I suggest clarifying time frame on the short-term.*

**A:** Thank you very much for your valuable comment. The "short-term" phrase refers to the period of CMPA (2008-2017) compared with the long period of the reconstruction data (1981-2017). In the revised manuscript, we changed the sentence to "Global 0.1° and local 0.25° *P* data in 1981-2017 are merged to reconstruct the historical P of the 0.1° China Merged Precipitation Analysis (CMPA) available in 2008-2017 using a stacking machine learning model." (Line 16-18)

**R2C5:** *4. On the names of the nine major river basins of China, I suggest replacing the Southwest Basin with the Southwest Basins, and replacing the Southeast Basin with the Southeast Basin, since either of them includes more than one basin.*

**A:** Thank you very much for your comments. A large basin consists of many sub-basins. The names "Southwest Basin" and "Southeast Basin" are used in the official boundary

data of nine major basins in China (https://www.resdc.cn/data.aspx?DATAID=141).
Therefore, we prefer to keep these names.